# Effect of MED-02 Containing Two Probiotic Strains, *Limosilactobacillus fermentum* MG4231 and MG4244, on Body Fat Reduction in Overweight or Obese Subjects: A Randomized, Multicenter, Double-Blind, Placebo-Controlled Study

**DOI:** 10.3390/nu14173583

**Published:** 2022-08-30

**Authors:** Young Gyu Cho, Yun Jun Yang, Yeong Sook Yoon, Eon Sook Lee, Jun Hyung Lee, Yulah Jeong, Chang Ho Kang

**Affiliations:** 1Department of Family Medicine, Seoul Paik Hospital, Inje University College of Medicine, Seoul 04551, Korea; 2Department of Family Medicine, Ilsan Paik Hospital, Inje University College of Medicine, Goyang 10380, Korea; 3MEDIOGEN, Co., Ltd., Biovalley 1-ro, Jecheon-si 27159, Korea

**Keywords:** overweight, obesity, probiotics, *Limosilactobacillus fermentum*, body fat mass, body fat percentage

## Abstract

MED-02 is a complex supplement containing two probiotic strains, *Limosilactobacillus fermentum* MG4231 and MG4244, isolated from humans. The anti-obesity effects and safety profile of MED-02 were assessed in overweight and obese subjects. In this randomized, double-blinded, placebo-controlled, multicenter study, 100 healthy obese and overweight subjects aged 19–65 years with a body mass index (BMI) between 25 and 31.9 kg/m^2^ were recruited and randomized to receive a placebo or MED-02 (5 × 10^9^ CFU/day). After 12 weeks of consumption, body fat mass (−1166.82 g vs. −382.08 g; *p =* 0.024) and body fat percentage (−0.85% vs. −0.11%; *p* = 0.030), as evaluated by dual-energy X-ray absorptiometry (DEXA) and body weight (−2.06 kg vs. −1.22 kg; *p* = 0.041), were significantly reduced in the MED-02 group compared to the placebo group. The safety profile did not differ among the groups. No serious adverse effects were observed in either group. These results suggest that MED-02 is a safe and beneficial probiotics that reduces body fat and body weight in overweight or obese individuals.

## 1. Introduction

Overweight and obesity are defined as excess body weight, which is a health risk owing to a high percentage of body fat [1]. According to the World Health Organization (WHO), more than 1.9 billion adults were overweight, about 650 million were obese, and 39% of the world’s population was already overweight or obese in 2016 [2]. Given the current trend, it is predicted that half of the world’s adult population will be overweight or obese by 2030; therefore, obesity is a serious problem worldwide. Obesity leads to various physical, psychological, and emotional problems, as well as social problems [3,4]. Obesity increases the risk of diseases, such as diabetes and hypertension, and the risk of the occurrence of and death from various cancers [5]. Therefore, proper management of obesity is required to improve quality of life, health, and lifespan.

There are several therapeutic modalities for obesity, including diet, exercise, behavioral therapy, surgery, and drugs [6]. In the treatment of obesity, it is difficult to induce sufficient weight loss in most obese patients using diet, exercise, and behavioral therapy alone; therefore, drug treatment is considered. Drugs can be used for both short and long periods [7]. However, these drugs may cause physical and psychological side effects, including gastrointestinal symptoms, hypertension, headache, depression, anxiety, insomnia, suicidal thoughts, and suicidal behavior. Therefore, there is an urgent need to find safe food or drugs applicable to obesity treatment with no side effects.

Probiotics are live microorganisms that, when ingested in adequate amounts, have a beneficial effect on the health of the host [8]. Recently, the effects of probiotics on the suppression of fat production, reduction of fat accumulation, triglyceride, and total cholesterol levels in plasma, and on inhibiting the lipolysis of fermented products have been reported [9,10,11,12]. *Limosilactobacillus fermentum* CECT5716 and *Lacticaseibacillus rhamnosus* GG alleviate obesity by modulating the gut microbiome in high-fat diet (HFD)-induced obese mice [13,14]. Two probiotic strains, *Latilactobacillus curvatus* HY7601 and *Lactiplantibacillus plantarum* KY1032, also reduced weight, fat adiposity, and lipoprotein-associated phospholipase A_2_ levels in overweight subjects in clinical trials [15].

MED-02 is a complex of the human-derived probiotic strains *L. fermentum* MG4231 and MG4244. *L. fermentum* is a probiotic species listed on the European QPS list and the FDA GRAS. In a previous study, *L. fermentum* MG4231 and MG4244 inhibited the differentiation and accumulation of triglycerides in 3T3-L1 pre-adipocytes [16]. In an HFD-induced obese mice model, the combination of *L. fermentum* MG4231 and MG4244 significantly reduced body weight, body fat, liver triglycerides, and fat glomerular size [17]. In addition, peroxisome proliferator-activated receptor γ (PPARγ), fatty acid synthase (FAS), lipoprotein lipase (LPL), and adipocyte protein 2 (aP2), which are related to adipogenesis, were significantly reduced by the activation of AMP-activated protein kinase (AMPK) and acetyl-CoA carboxylase (ACC), which promote fat oxidation and inhibit fat synthesis.

Although there are several studies supporting the anti-obesity effect of MED-02, evaluating the effects of MED-02 on human body fat is necessary.

In this study, we evaluated body fat reduction caused by the ingestion of the MED-02 probiotic complex containing *L. fermentum* MG4231 and MG4244 in non-diabetic and overweight individuals.

## 2. Materials and Methods

### 2.1. Study Design and Treatment Materials

This study was a 12-week, randomized, double-blind, placebo-controlled, multi-center clinical study performed at Seoul Paik Hospital, Seoul, Korea and Ilsan Paik Hospital, Goyang, Korea. The study protocol and consent form were approved by the independent clinical trial review board (IRB) of each institution (IRB No. PAIK 2020-09-010-002 and ISPAIK 2020-09-033-001). This study was registered with the Clinical Research Information Service (CRiS), Republic of Korea (KCT0005861, https://cris.nih.go.kr/cris/search/detailSearch.do/18211, accessed on 3 February 2021). The participants were instructed to take 500 mg of MED-02 or a placebo capsule with water once daily for 12 weeks. The MED-02 probiotic capsule contained two probiotic strains, *L. fermentum* MG4231 and MG4244, each at 2.5 × 10^9^ CFU, with 250 mg of maltodextrin and a placebo capsule containing maltodextrin only. All participants visited the hospital at 6 and 12 weeks for the assessment of anthropometric parameters after receiving the investigated products.

### 2.2. Subjects

The study subjects were recruited from Seoul Paik Hospital, Seoul, Korea, and Ilsan Paik Hospital, Goyang, Korea. Healthy obese and overweight male and female subjects aged 19–65 years with body mass index (BMI) ranging from 25.0 to 31.9 kg/m^2^ at screening and baseline were recruited for this study. The participants were informed of the purpose and protocol of the study and the foreseeable risks involved in the trial. All the participants voluntarily signed a written informed consent form. The subjects were evaluated for eligibility according to the exclusion criteria, as follows: receiving treatment for serious diseases (e.g., cardiovascular, immunological, respiratory, hepatobiliary, renal, neurological, musculoskeletal, mental, and infectious diseases, or malignant tumor); taking any bariatric drug for weight loss (e.g., anorexiants, fat absorption inhibitors, glucagon-like peptide-1 (GLP-1) receptor agonists), psychiatric drugs for depression and schizophrenia, β-blockers, diuretics, contraceptives, steroids, female hormonal injections, or additional health supplements for weight control within one month of screening visit; taking any antibiotic drug within two weeks of screening visit; consuming any probiotic product within a month of screening; blood pressure over 160/100 mmHg after a 10-min rest; fasting plasma glucose levels over 126 mg/dL or diabetic patients taking antidiabetic drugs (insulin, hypoglycemic agents); serum thyroid-stimulating hormone (TSH) levels below 0.1 μIU/mL or above 10 μIU/mL; serum creatinine levels two times higher than the normal standard; serum aspartate aminotransferase (AST; GOT) or alanine aminotransferase (ALT; GPT) levels three times higher than the normal standard; severe gastrointestinal disorder; subjects had been hospitalized, treated with drugs, or rehabilitated because of alcohol use, alcohol-induced disorders, heart disease, or central nervous system disorders; suffering from musculoskeletal disorder; subjects had a weight change of 10% or more within three months of screening visit; subjects had participated in a commercial obesity program within three months of screening visit; subjects had participated or were planning to participate in other interventional clinical trials (including human clinical trials) within three months of screening visit; women who had undergone, or planned to undergo, pregnancy or nursing during the trials; sensitive or allergic to ingredients included in the test formulation; and subjects were considered to be inappropriate to participate in the study by the investigator.

A total of 105 obese or overweight applicants were enrolled, and 100 subjects were randomly assigned to the placebo and MED-02 groups in a 1:1 ratio at baseline (50 participants were allocated to each group).

### 2.3. Assessment of Daily Energy Intake and Physical Activity

During the study period, all subjects in both groups were recommended to reduce their energy intake by 500 kcal/day less than usual and to burn 300 kcal or more by exercising every day. A standardized 3-day dietary record (2 weekdays and 1 weekend day before baseline, at 6 weeks, and at 12 weeks) was completed for each participant at home after they had received detailed instructions from a dietitian. A computerized version of the Korean Nutrition File (Can-Pro 4.0; The Korean Nutrition Society, Seoul, Korea) was used to determine the macronutrient content of foods and the total daily energy intake. Physical activity was recorded using the Global Physical Activity Questionnaire (GPAQ) at baseline, 6 weeks, and 12 weeks and then quantified by metabolic equivalent of task (MET) value as follows:MET value (min/week) = (V × 8) + (M × 4) + (T × 4)(1)
where V is the vigorous-intensity activity that causes large increases in breathing or heart rate for at least 10 min continuously during work and recreational time, M is the moderate-intensity activity that causes small increases in breathing or heart rate for at least 10 min continuously during work and recreational time, and T is the time for walking or riding bicycle to move places. The participants were requested to record smoking and alcohol drinking habits at baseline, 6 weeks, and 12 weeks.

### 2.4. Clinical Outcomes

The primary efficacy outcomes were a reduction in body fat mass and percentage from baseline to 12 weeks after randomization. Body fat mass and percentage were analyzed using dual-energy X-ray absorptiometry (DEXA). The secondary efficacy outcomes were changes in body weight, waist and hip circumferences, waist-to-hip ratio, BMI, lean body mass, visceral and subcutaneous fat areas, total abdominal fat area, visceral-to-subcutaneous ratio, blood lipids (e.g., total cholesterol, low-density lipoprotein (LDL) cholesterol, high-density lipoprotein (HDL) cholesterol, and triglycerides), adiponectin, and leptin. The visceral and subcutaneous fat areas and total abdominal fat area were measured from the L4–L5 intervertebral space using a computerized tomography (CT) scanner. Blood lipids were evaluated with fasting blood samples with the enzymatic colorimetric method at Seoul Paik Hospital and Ilsan Paik Hospital. Adipokines were evaluated with ELISA kits. Safety was assessed by monitoring all adverse events and results from the blood chemical tests (e.g., ALT, AST, and γ-glutamyl transpeptidase (γ-GTP) for the liver function tests; blood urea nitrogen (BUN) and creatinine for the renal function tests; and high-sensitivity C-reactive protein (hs-CRP) for the inflammatory status) and vital signs, blood pressure, and pulse. Adverse events were found by non-directive questioning or voluntary reporting from participants.

### 2.5. Statistical Analysis

Statistical analyses were performed using SAS^®^ (version 9.4; SAS Institute, Cary, NC, USA). The full analysis and per-protocol datasets were used for statistical analyses. Significant differences in demographic characteristics at baseline were analyzed using the Chi-square test, Fisher’s exact test, Wilcoxon rank sum test, or two-sample *t*-test. The significance of the differences in the changes between baseline and at 12 weeks within groups was analyzed using a paired *t*-test, and the differences in the changes from baseline to 12 weeks between the placebo and MED-02 groups were analyzed using a two-sample *t*-test or Wilcoxon rank sum test. In addition, analysis of covariance (ANCOVA) was performed using a generalized linear model (GLM) analysis with sex, age, drinking, smoking, exercise, and site as covariates. A GLM was planned to be conducted if there was any significant difference between the base characteristic groups, but there was no statistically significant difference among the base characteristic groups. Data are presented as mean ± standard deviation (SD). Differences were considered statistically significant at a *p*-value < 0.05.

## 3. Results

### 3.1. Baseline Characteristics of Subjects

In this trial, 105 subjects were screened before a total of 100 subjects were randomly assigned to the MED-02 group (*n =* 50) or placebo group (*n =* 50). Among them, 7 of 100 subjects were excluded because of the occurrence of moderate adverse reactions (not related to MED-02) or COVID-19, failure to visit at 12 weeks, withdrawal of consent, violation of the selection/exclusion criteria, and failure to follow the standard protocol. Additionally, there was one more violation according to the selection/exclusion criteria after completing intake. Therefore, 45 subjects in the MED-02 group and 47 subjects in the placebo group (92 subjects) were administered the test and control products, respectively. Thereafter, the efficacy of MED-02 was evaluated in 38 subjects in the MED-02 group and 37 subjects in the placebo group without significant violations affecting the test results (Figure 1, four subjects violated the meeting date, three subjects ingested synthroid, two subjects ingested Oriental medicine, one subject ingested an Oriental supplement, one subject ingested a prohibited supplement, two subjects ingested prohibited drugs, and four subjects followed very low-calorie diets for the entire dietary recordings).

Table 1 shows the differences in demographic information and characteristics data, including work and exercise, based on the global physical activity questionnaire and evaluation. There were 21 male participants (55.26%) in the MED-02 group and 24 (64.86%) in the placebo group (*p* = 0.396, chi-square test). The mean values of age were 43.34 ± 9.94 years and 44.59 ± 9.77 years in the MED-02 group and placebo group, respectively (*p =* 0.584 by two-sampled *t*-test). In addition, there were no statistically significant differences between the two intake groups in alcohol drinking, frequency, and quantity, duration of smoking, and physical activity (working, walking to move places, or exercising); therefore, comparability between groups could be assumed.

### 3.2. Effect of MED-02 on Body Fat Mass and Percentage

When probiotics were ingested for 12 weeks, changes in body fat mass and percentage were analyzed as primary efficacy evaluation variables. As a result, body fat mass was significantly decreased by 1166.82 ± 1741.13 g (range: −6702.00~1061.00 g; *p <* 0.001) in the MED-02 group and 382.08 ± 1284.46 g (−4064.00~2319.00 g; *p =* 0.079) in the placebo group. There was a statistically significant difference in the change in body fat mass after 12 weeks between the two intake groups (Figure 2A,B; *p =* 0.024). In the analysis of body fat percentage, the fat percentage significantly decreased by 0.85 ± 1.57% (−5.60~1.70%; *p =* 0.002) in the MED-02 group and 0.11 ± 1.35% (−4.30~3.30%; *p =* 0.621) in the placebo group, indicating a statistically significant difference between the two intake groups (Figure 2C,D; *p =* 0.030). The lean body mass, a secondary significance evaluation variable, decreased by 403.05 ± 1313.36 g (−5137.00~1829.00 g; *p =* 0.066) and 459.41 ± 1117.33 g (−2553.00~1533.00 g; *p =* 0.017) in the MED-02 and placebo administrated groups, respectively, after 12 weeks of ingestion, but there was no statistically significant difference between the two intake groups (Table 2). The maximum changes in body fat mass and percentage were −6702.00 g and −5.6%, respectively, when MED-02 was administered for 12 weeks.

### 3.3. Effect of MED-02 on Body Weight and BMI

Body weight and BMIs were measured after 6 and 12 weeks of administration. At the baseline, there was no significant difference in body weight between the two intake groups. After 6 weeks of ingestion, body weight significantly decreased by 1.11 ± 1.87 kg (−6.50~2.30 kg; *p <* 0.001) in the MED-02 group and 0.43 ± 1.43 kg (−5.60~2.00 kg; *p =* 0.076) in the placebo group. A statistically significant difference was observed in body weight change after 6 weeks between the two intake groups (*p =* 0.042). After 12 weeks, body weights decreased by 2.06 ± 2.21 kg (−9.50~0.70 kg; *p <* 0.001) and 1.22 ± 1.73 kg (−5.70~1.90 kg; *p <* 0.001) in the MED-02 and placebo groups, respectively, indicating a statistically significant difference between the MED-02 and placebo groups (Figure 2E,F; *p =* 0.041). After 12 weeks, BMI values significantly decreased by 0.70 ± 0.73 kg/m^2^ (−2.90~0.20 kg/m^2^; *p <* 0.001) and 0.44 ± 0.60 kg/m^2^ (−2.00~0.60 kg/m^2^; *p <* 0.001) in the MED-02 and placebo groups, respectively. There was no statistical difference in the change in BMI after 12 weeks between the two intake groups (*p =* 0.064).

### 3.4. Effect of MED-02 on Abdominal Fat Area

The magnitude of change in the abdominal fat area after 12 weeks of intake was evaluated using a CT scan (Table 2). At the baseline, there was no significant difference between the two intake groups. Visceral fat areas were significantly reduced by 11.43 ± 24.25 cm^2^ (−81.50~27.00 cm^2^; *p =* 0.006) in the MED-02 group and 5.78 ± 25.06 cm^2^ (−85.00~67.00 cm^2^; *p =* 0.169) in the placebo group. However, there was no statistically significant difference in the changes in visceral fat areas after 12 weeks between the two intake groups (*p =* 0.304). In addition, changes in other variables, such as subcutaneous fat areas, total abdominal fat area, and visceral/subcutaneous fat area ratio, after 12 weeks were not statistically significant in difference between the two intake groups.

### 3.5. Effect of MED-02 on the Anthropometric Parameters and the Plasma Levels of Lipid Metabolism Markers and Adipokines

As shown in Table 2, there was no significant difference at the baseline between the two intake groups. Waist and hip circumferences significantly decreased after 12 weeks of intake in both the MED-02 and placebo groups (*p <* 0.001, all), but there was no statistically significant difference in changes in waist and hip circumferences after 12 weeks between the two intake groups (*p =* 0.594 and 0.290, respectively). Waist/hip ratio circumferences decreased in both the MED-02 and placebo groups (*p =* 0.176 and 0.015, respectively), which was not a statistically significant difference between the groups (*p =* 0.750).

The results of plasma analysis showed a reduction in the levels of total cholesterol and LDL-cholesterol in the MED-02 group (*p* = 0.491 and 0.885, respectively) and an increase in the placebo group (*p* = 0.490 and 0.459, respectively) after 12 weeks. Additionally, the levels of HDL-cholesterol, triglyceride, and adiponectin decreased, and the level of leptin increased in both groups. However, changes in lipid metabolism markers and adipokines after 12 weeks were not statistically significant in difference between the two intake groups.

### 3.6. Assessment of the Safety of MED-02

To evaluate the safety of MED-02, plasma levels of glucose, AST, ALT, total bilirubin, ALP, RBC, Hb, Hct, WBC, platelets, lymphocytes, creatinine, BUN, uric acid, γ-GTP, glucose, systolic and diastolic blood pressure, and pulse rates were measured (Table 3). At the baseline, there was significant difference in AST, ALT, ALP, and RBC between the two intake groups. However, all the parameters were within the normal ranges and did not change significantly between the two intake groups. This indicates that daily intake of MED-02 for 12 weeks did not cause any health problems. During each visit, the subjects were evaluated for side effects or symptoms. There were 11 adverse events that occurred in the MED-02 group and 14 adverse events that occurred in the placebo group, but there were no serious adverse events. There was no significant difference in adverse events between the two intake groups (*p* = 0.806). The most common adverse events were skin reactions, such as eczema, dermatitis, urticaria, alopecia, and verruca, which accounted for six adverse events (three in the MED-02 group and three in the placebo group). In addition, there were five cases of musculoskeletal symptoms (two in the MED-02 group and three in the placebo group). Three cases of general symptoms, such as back pain and fever, one case of a peptic ulcer, one case of a breast benign neoplasm, and one case of rhinitis were reported only in the MED-02 group and two cases of headache, one case of sialadenitis, one case of periodontal disease, one case of increased hepatic enzymes, one case of increased creatine phosphokinase, one case of benign prostatic hyperplasia, and one case of keratoconjunctivitis were reported only in the placebo group.

## 4. Discussion

This study investigated the anti-obesity effects of MED-02, a complex of *Limosilactobacillus fermentum* MG4231 and MG4244, isolated from humans. A reduction in body fat mass and body fat present was determined by DEXA and CT scanning after 12 weeks of administration via randomized, double-blinded, multi-center, placebo-controlled clinical trials in obese or overweight Korean adults aged 19 to 65 years with a BMI ranging from 25.0 kg/m^2^ to <32.0 kg/m^2^.

There were statistically significant differences between the MED-02 and placebo groups in changes in body fat mass, body fat percentage, and body weight after 12 weeks (*p =* 0.024, 0.030, and 0.041, respectively). In addition, waist circumference, hip circumference, BMI, visceral fat area, and abdominal fat in the MED-02 group decreased significantly compared to the baseline (*p <* 0.001, *p <* 0.001, *p <* 0.001, *p =* 0.006, and *p =* 0.003, respectively). In the safety evaluation, there was no difference between the MED-02 group and the placebo group in terms of adverse reactions and serious adverse events, and there was no statistically significant difference between the groups in the clinical pathology results, such as hematology, blood chemistry, urinalysis, and vital signs.

Similarly, in the study of *Lactilobacillus sakei* CJLS03 (5 × 10^9^ CFU/day), a 12-week human application study targeting overweight and obese adults aged 20–65 with a BMI over 25 kg/m^2^, there were significant differences in body fat mass, waist circumference, and abdominal visceral fat between the probiotics and placebo intake groups (*p* < 0.05) [18]. According to Minami et al. [19], adults aged 20–64 years with BMIs of 25–30 kg/m^2^ ingested *Bifidiobacterium breve* B-3 (2 × 10^10^ CFU/day) for 12 weeks. The changes in body fat mass and percentage were significantly more decreased in the *B. breve* B-3 ingesting group than in the placebo group (*p <* 0.05). In addition, the levels of triglycerides and HDL-cholesterol were slightly improved after probiotic intake (*p <* 0.1) [19].

There are several mechanisms by which probiotics reduce adiposity [15]. The metabolites of the two probiotic strains may directly affect body fat composition. Body weight and adipose tissue weight were significantly decreased after feeding mice MED-02 in an HFD-induced obese mice model [17]. The expression of pAMPK in epididymal fat was increased three-fold in the negative placebo group; thus, the expression of pACC, leading to fat oxidation, was increased. In addition, as the pAMPK expression level increased, the expression of PPARγ, which regulates the maturation of preadipocytes and the expression of lipogenesis-related factors (FAS, aP2, and LPL) significantly decreased. Therefore, MED-02 consumption has an effect on lipid synthesis and accumulation in HFD-induced obese mice.

Another possible theory is the modulation of the gut microbiome by the consumption of probiotics. Recently, it was reported that the gut microbiota plays an important role in energy metabolism and nutrient absorption in the human body [20]. The composition of the gut microbiome in an obese body is completely different from that of a lean body [10,21]. The gut microbiome of obese individuals showed an increase in the microbial community of Firmicutes and a decrease in the phylum Bacteroidetes compared to that of a lean body [22,23]. In addition, compared to subjects with an abundant microbiome, subjects with low richness in the gut microbiome had higher levels of C-reactive protein, leptin, dyslipidemia, insulin resistance, and inflammatory phenotypes, and gained more body weight and body fat [24]. It is not fully understood how the gut microbiome is related to obesity and metabolic syndrome, but many studies have confirmed that it effectively alleviates obesity after the consumption of probiotics [10,11].

In this study, the body weight, body fat mass, and body fat percentages of the MED-02 group decreased to a greater extent than those of the placebo group. These results are effective and safe without any side effects that are common in chemical medications. With the development of analytical technology, various methods have been proposed to study the effect of probiotics on the intestinal microflora for the alleviation of obesity [20]. However, we did not evaluate changes in the gut microbiota in this study. Therefore, further studies are needed to investigate the correlation between the reduction in body weight and gut microbiota in humans to understand the anti-obesity effect of MED-02. In addition, MED-02 probiotics modulate obesity through the AMPK system, but it needs to be made clear which components from probiotics modulate the system in further studies.

## 5. Conclusions

In conclusion, MED-02, a mixture of *L. fermentum* MG4231 and MG4244 (1:1), reduced body fat mass, body fat percentage, and body weight in overweight or obese subjects without side effects. The findings in this study are supported by previous animal experiments. Altogether, our results show that MED-02 may have beneficial effects in overweight and obese adults.

## Figures and Tables

**Figure 1 nutrients-14-03583-f001:**
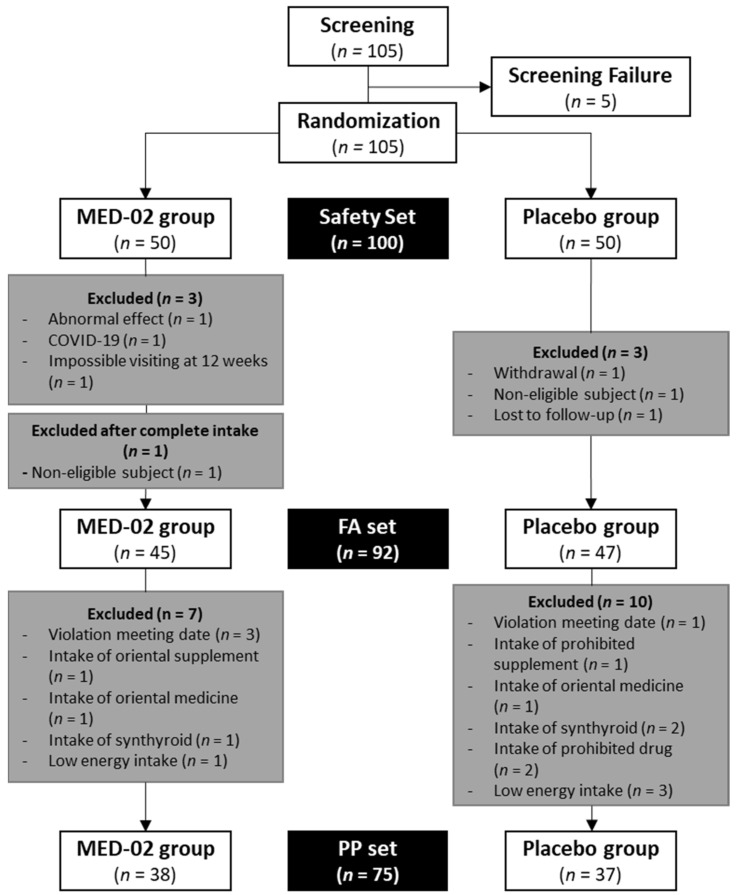
Flow diagram of the enrolled participants.

**Figure 2 nutrients-14-03583-f002:**
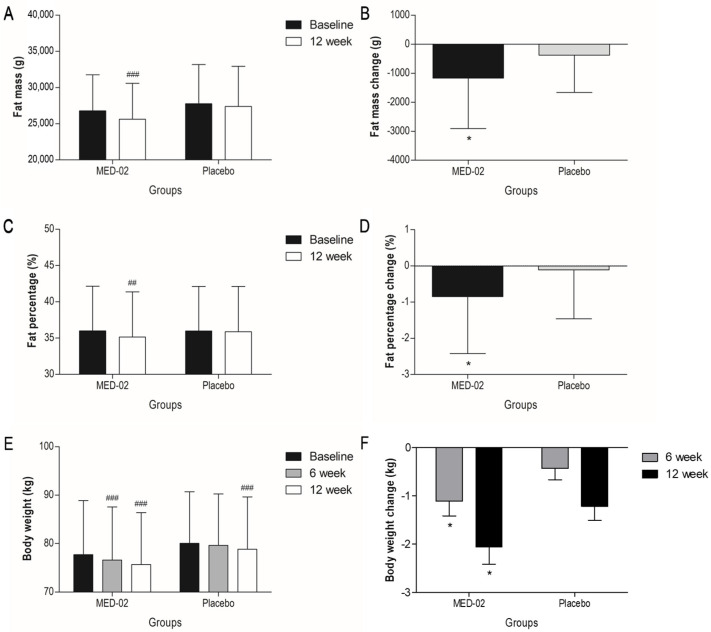
Changes in (**A**,**B**) body fat mass, (**C**,**D**) body fat percentage in the MED-02 and placebo groups were evaluated by DEXA at baseline and 12 weeks. (**E**,**F**) body weight changes in the MED-02 and placebo groups were evaluated at baseline, 6 weeks, and 12 weeks. There was no significant difference at the baseline between two intake groups. Values are presented as mean ± SD; ## *p <* 0.01 and ### *p <* 0.001 compared with baseline in each group by paired *t*-test; * *p <* 0.05 compared between groups by ANCOVA adjusted for sex, age, drinking, smoking, exercise, and site.

**Table 1 nutrients-14-03583-t001:** Baseline demographic characteristics of the subjects.

Variable	MED-02 (*n =* 38)	Placebo (*n =* 37)	*p*-Value
Sex (M/F)	21/17	24/13	0.396 ^C^
Age	43.34 ± 9.94	44.59 ± 9.77	0.584 ^T^
Drinker (Y/N)	24/14	24/13	0.924 ^F^
Smoker (Y/N)	6/32	9/28	0.848 ^F^
Physical activity (MET-min/week)	4647.65 ± 4767.41	4980.59 ± 5468.15	0.980 ^W^
Energy intake (kcal/day)	1482.51 ± 431.51	1610.41 ± 473.68	0.225 ^T^
Carbohydrate (g) % of energy intake	201.22 ± 69.43 55.39 ±11.46	211.13 ± 72.80 54.37 ± 10.75	0.548 ^T^ 0.870 ^T^
Protein (g) % of energy intake	64.73 ± 23.11 17.11 ± 4.33	67.01 ± 24.29 16.42 ± 3.47	0.678 ^T^ 0.394 ^T^
Fat (g) % of energy intake	44.04 ± 19.57 23.75 ± 8.69	49.55 ± 17.36 24.32 ± 5.88	0.201 ^T^ 0.233 ^T^
Fiber (g)	15.46 ± 5.89	15.89 ± 5.44	0.740 ^T^

Values are presented as mean ± SD; *p*-values are analyzed by ^C^ Chi-square test, ^T^ two-sample *t*-test, ^F^ Fisher’s exact test, or ^W^ Wilcoxon rank sum test; MET, metabolic equivalent of task; M/F, male/female; Y/N, Yes/No; EI, energy intake.

**Table 2 nutrients-14-03583-t002:** Anthropometric parameters and blood chemistry of subjects before and after 12-week placebo and consumption of probiotics (MED-02).

Variable	MED-02 (*n =* 38)			*p*- Value ^a^	Placebo (*n =* 37)			*p*- Value ^a^	*p*- Value ^b^
	Baseline	12 Weeks	Change (12 Weeks–Baseline)		Baseline	12 Weeks	Change (12 Weeks–Baseline)		
**Anthropometric parameters**									
BMI (kg/m^2^)	27.27 ± 1.91	26.57 ± 2.05	−0.70 ± 0.73	<0.001	28.15 ± 1.99	27.71 ± 1.98	−0.44 ± 0.60	<0.001	0.064
Waist (cm)	90.08 ± 7.93	87.78 ± 7.50	−2.30 ± 2.29	<0.001	92.42 ± 7.45	90.38 ± 7.35	−2.04 ± 2.43	<0.001	0.594
Hip (cm)	101.63 ± 4.63	99.45 ± 4.22	−2.18 ± 1.83	<0.001	102.68 ± 4.89	101.22 ± 4.98	−1.46 ± 1.95	<0.001	0.290
Waist-to-hip ratio	0.89 ± 0.06	0.88 ± 0.05	−0.00 ± 0.02	0.176	0.90 ± 0.05	0.89 ± 0.05	−0.01 ± 0.02	0.015	0.750
**DEXA measurement**									
Lean body mass (g)	47,111.29 ± 8935.15	46,708.24 ± 8574.14	−403.05 ± 1313.36	0.066	48,670.22 ± 8625.62	48,210.81 ± 8725.14	−459.41 ± 1117.33	0.017	0.590
**CT measurement**									
Visceral fat area (cm^2^)	112.59 ± 48.49	101.16 ± 40.72	−11.43 ± 24.25	0.006	132.64 ± 55.79	126.86 ± 53.79	−5.78 ± 25.06	0.169	0.304
Subcutaneous fat area (cm^2^)	219.44 ± 66.68	211.14 ± 66.32	−8.29 ± 31.09	0.109	222.93 ± 61.56	212.26 ± 66.88	−10.67 ± 33.29	0.059	0.538
Total abdominal fat area (cm^2^)	332.03 ± 86.32	312.30 ± 85.89	−19.73 ± 38.66	0.003	355.57 ± 84.34	339.12 ± 88.29	−16.45 ± 36.86	0.010	0.873
Visceral-to-subcutaneous fat ratio	0.55 ± 0.30	0.51 ± 0.23	−0.04 ± 0.16	0.117	0.64 ± 0.31	0.65 ± 0.31	0.01 ± 0.16	0.760	0.149
**Blood chemistry**									
Total cholesterol (mg/dL)	200.00 ± 29.89	197.79 ± 31.92	−2.21 ± 19.60	0.491	201.89 ± 30.17	204.45 ± 31.10	2.55 ± 22.30	0.490	0.254
HDL-cholesterol (mg/dL)	56.16 ± 12.61	54.82 ± 11.03	−1.34 ± 7.02	0.246	54.51 ± 11.96	53.97 ± 10.71	−0.54 ± 6.04	0.589	0.989
LDL-cholesterol (mg/dL)	125.79 ± 25.09	125.37 ± 29.06	−0.42 ± 17.77	0.885	126.95 ± 25.73	129.22 ± 22.86	2.27 ± 18.46	0.459	0.375
Triglyceride (mg/dL)	132.68 ± 89.59	118.39 ± 58.60	−14.29 ± 51.88	0.098	156.57 ± 99.33	147.08 ± 78.09	−9.49 ± 80.63	0.479	0.862
hs-CRP	0.15 ± 0.27	0.10 ± 0.10	−0.05 ± 0.28	0.283	0.15 ± 0.22	0.14 ± 0.14	−0.01 ± 0.18	0.740	0.611
Adiponectin (µg/mL)	11.27 ± 7.37	9.38 ± 4.97	−1.89 ± 4.43	0.012	9.06 ± 5.16	7.81 ± 3.98	−1.25 ± 2.48	0.004	0.561
Leptin (ng/mL)	13.07 ± 7.80	13.32 ± 8.31	0.25 ± 5.95	0.796	13.55 ± 8.03	14.47 ± 7.77	0.92 ± 4.48	0.221	0.500

Values are presented as mean ± SD; ^a^ compared within groups, *p*-values were analyzed by paired *t*-test; ^b^ compared between groups, *p*-values were analyzed by ANCOVA adjusted for sex, age, drinking, smoking, exercise, and site.

**Table 3 nutrients-14-03583-t003:** Safety assessment of subjects before and after 12-week consumption of probiotics.

Variable	MED-02 (*n =* 50)		Placebo (*n =* 50)		*p*-Value
	Baseline	12 Weeks	Baseline	12 Weeks	
AST (IU/L)	22.40 ± 5.74	21.33 ± 5.72	26.06 ± 8.03	26.04 ± 14.86	0.439 ^W^
ALT (IU/L)	21.14 ± 11.48	19.74 ± 11.62	31.06 ± 21.28	30.53 ± 19.76	0.930 ^W^
ALP (IU/L)	62.28 ± 12.72	71.48 ± 16.36	62.13 ± 12.48	72.23 ± 16.06	0.603
γ-GTP	26.06 ± 27.66	20.91 ± 14.80	32.82 ± 27.96	32.40 ± 27.73	0.464 ^W^
Glucose (mg/dL)	102.48 ± 10.03	100.93 ± 8.87	101.76 ± 8.69	100.47 ± 9.66	0.976
Total bilirubin (mg/dL)	0.77 ± 0.38	0.76 ± 0.28	0.74 ± 0.36	0.77 ± 0.34	0.360
Creatinine (mg/dL)	0.76 ± 0.16	0.77 ± 0.17	0.80 ± 0.18	0.82 ± 0.20	0.445
BUN (mg/dL)	13.96 ± 4.24	13.18 ± 2.96 *	13.74 ± 3.86	13.34 ± 3.01	0.797 ^W^
Uric acid (mg/dL)	5.69 ± 1.20	5.62 ± 1.26	6.23 ± 1.52	6.29 ± 1.68	0.576
RBC (10^6^/µL)	4.80 ± 0.47	4.78 ± 0.47	5.02 ± 0.54	4.99 ± 0.56	0.648 ^W^
Hb (g/dL)	14.31 ± 1.49	14.18 ± 1.50	14.73 ± 1.65	14.80 ± 1.54	0.529
Hct (%)	42.97 ± 3.74	42.63 ± 3.99	44.16 ± 4.12	44.17 ± 4.19	0.737
WBC (10^3^/µL)	6.11 ± 1.57	6.04 ± 1.30	6.46 ± 1.41	6.38 ± 1.27	0.727 ^W^
Platelet (10^3^/µL)	264.00 ± 43.73	263.80 ± 42.63	273.28 ± 52.20	272.66 ± 54.58	0.560
Neutrophil (%)	54.81 ± 8.99	53.94 ± 7.78	54.69 ± 8.65	53.15 ± 8.21	0.990
Lymphocyte (%)	35.00 ± 7.99	35.81 ± 7.22	35.36 ± 8.06	37.02 ± 7.62	0.768
Monocyte (%)	6.75 ± 1.64	6.90 ± 1.46	6.90 ± 1.60	7.00 ± 1.54	0.895
Systolic blood pressure (mmHg)	132.38 ± 12.26	128.35 ± 7.88 *	135.00 ± 11.42	129.79 ± 12.06 *	0.687
Diastolic blood pressure (mmHg)	81.58 ± 8.02	79.63 ± 6.56	82.56 ± 7.38	80.19 ± 8.66	0.814
Pulse (times/min)	78.42 ± 9.50	74.20 ± 9.51 *	78.76 ± 11.66	76.49 ± 10.64	0.482

Values are presented as mean ± SD; * *p* < 0.05 derived from statistical analysis within groups; *p*-values were analyzed by two-sample *t*-test or ^W^ Wilcoxon rank sum test in changed values between groups; AST, aspartate aminotransferase; ALT, alanine aminotransferase; ALP, alkaline phosphatase; Hb, hemoglobin; Hct, hematocrit; RBC, red blood cell; WBC, white blood cell; BUN, blood urea nitrogen; γ-GTP, gamma-glutamic transpeptidase.

## Data Availability

Not applicable.

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
