# Peer review of "Effect of MED-02 Containing Two Probiotic Strains, Limosilactobacillus fermentum MG4231 and MG4244, on Body Fat Reduction in Overweight or Obese Subjects: A Randomized, Multicenter, Double-Blind, Placebo-Controlled Study"

_nutrients, 2022, doi:10.3390/nu14173583_

Round 1
Reviewer 1 Report
1, In the method part, please report which questionnaire used for symptom or toxicity assessment. Like CTCAE? In 3.6. Assessment of the safety of MED-02, please also report symptom (especially gastrointestinal toxicity) and corresponding degree apart from lab results. The description “No severe side effects were observed during the intervention” is too obscure.
2, For Fig 2F, as the data of 6 weak is available, it is recommended that BW change of baseline-6 weak is also plotted.
3, For Fig 2E, according to your results below:
“After 6 weeks of ingestion, the body weight significantly decreased by 1.11 ± 1.87 kg (p <
0.001) in the MED-02 group, After 12 weeks, the body weights decreased by 2.06 ± 2.21 kg (p < 0.001) and 1.22 ± 1.73 kg (p < 0.001) in the MED-02”
Why “ ###p <0.001” is not highlighted in Fig 2E? Please check the figures again.
4. For dietary assessment, only total energy intake is reported, but main sources of energy intake should also be considered. Fiber is also essential factor to be considered. Thus, factors below can be reassessed and added cording to your 3-day dietary record:
Carbohydrate (g) % of Energy intake
Protein (g) % of Energy intake
Fat (g) % of Energy intake
Fiber (g)
5, Some minor mistakes concerned:
Page 2:
1) high fat induced mice should be high fat induced obese mice
2) obesity-induced animal model should be high-fat diet-induced obesity in animal models
page 3
3) uncontrolled hypertensive should be hypertension
4) 25 to 31.9 kg/m2
Page 10
5) “BMI of 25 kg/cm2, more than 25 kg/cm2 and less than 32 kg/cm2”, why the cutoff is so strange? 25.0 to <30 for overweight, 30.0 to <32 for obesity is more common.
Page 11
6) Please indite the full name of “HFD”.
7) Animal trials should be animal experiments
Author Response
Response to Reviewer 1 Comments
We thank you for your constructive comments. The Reviewer’s comments and our responses to them are described below. The original comments by the reviewer are in black font; they are followed by our responses in red font. Line numbers in our responses are the line number of the revised manuscript.
Point 1: In the method part, please report which questionnaire used for symptom or toxicity assessment. Like CTCAE? In 3.6. Assessment of the safety of MED-02, please also report symptom (especially gastrointestinal toxicity) and corresponding degree apart from lab results. The description “No severe side effects were observed during the intervention” is too obscure.
Response 1: Thank you for your comments. Information on adverse reactions was searched by non-directive questioning or voluntary reporting at visits. Also, the information was reported through blood chemistry or urinary examinations. The method was added in the line 154-155 and the results were edited in line 274-288.
Point 2: For Fig 2F, as the data of 6 weak is available, it is recommended that BW change of baseline-6 weak is also plotted.
Response 2: Thank you for your recommendation. We added BW change of 6 weeks data in Fig. 2F.
Point 3: For Fig 2E, according to your results below:
“After 6 weeks of ingestion, the body weight significantly decreased by 1.11 ± 1.87 kg (p <
0.001) in the MED-02 group. After 12 weeks, the body weights decreased by 2.06 ± 2.21 kg (p < 0.001) and 1.22 ± 1.73 kg (p < 0.001) in the MED-02.”.
Why “ ###p <0.001” is not highlighted in Fig 2E? Please check the figures again.
Response 3: There was a mistake. The p-value data was corrected in Fig. 2E.
Point 4: For dietary assessment, only total energy intake is reported, but main sources of energy intake should also be considered. Fiber is also essential factor to be considered. Thus, factors below can be reassessed and added cording to your 3-day dietary record:
Carbohydrate (g) % of Energy intake
Protein (g) % of Energy intake
Fat (g) % of Energy intake
Fiber (g)
Response 4: Thank you for your advice. As your comment, the data is added in table 1.
Point 5: high fat induced mice should be high fat induced obese mice
Response 5: Thank you for your comment. As your comment, we added obese in line 55.
Point 6: obesity-induced animal model should be high-fat diet-induced obesity in animal models
Response 6: Thank you for your comment. As your comment, we corrected the words in line 61-62.
Point 7: uncontrolled hypertensive should be hypertension
Response 7: Thank you for your comment. We corrected the word as “blood pressure over 160/100 mmHg” in line 102.
Point 8: 25 to 31.9 kg/m2
Response 8: Thank you for your comment. The unit of BMI on this paper was corrected as kg/m2.
Point 9: “BMI of 25 kg/cm2, more than 25 kg/cm2 and less than 32 kg/cm2”, why the cutoff is so strange? 25.0 to <30 for overweight, 30.0 to <32 for obesity is more common.
Response 9: Thank you for your comment. The product, MED-02 probiotics, used in this study is a food supplement with a potential to reduce body fat, not an anti-obesity drug. In the case of morbidly obese patients, it may be an ethical issue to participate in a clinical trial evaluating the body fat reducing effect of a food supplement that have not yet been verified without using an appropriate anti-obesity drug. Although overweight subjects with a BMI of 25 to 30 kg/m2 were the main subjects in the clinical trial, we included slightly obese subjects with a BMI of 30 to 32 in the trial to evaluate the effect of MED-02 in subjects with a range of obesity.
Point 10: Please indite the full name of “HFD”.
Response 10: Thank you for your comment. The full name of HFD was indited in line 54.
Point 11: Animal trials should be animal experiments
Response 11: Thank you for your comment. The word was corrected in line 357.
Reviewer 2 Report
The analysis of probiotic effects on some diseases like obesity provides scientific knowledge supporting its clinical use and, therefore, is a valuable input. The authors suggested that MED-02 could be a potential probiotic supplement to prevent obesity.
In general terms, it is an interesting study, but some questions need to be addressed:
- Materials and Methods
Study Design and Treatment Materials
The authors should describe the capsule composition with the administered probiotic and the placebo in detail.
Assessment of daily energy intake and physical activity
The authors mentioned, “During the study period, the subjects were recommended to reduce their energy in-take by 500 kcal/day less than usual and to burn 300 kcal or more by exercising every day”.
Why did they make that recommendation? In this way, the effect of the probiotic cannot be evaluated because it could be overlapped with dietary caloric restriction.
Which was the daily Kcal uptaken of the participants? Also, were participants monitored to maintain those daily calories until the end of the study?
Clinical Outcomes
The authors should mention the methodology for blood lipids determinations (total cholesterol, low-density lipoprotein (LDL)- cholesterol, high-density lipoprotein (HDL)-cholesterol, and triglycerides), adiponectin, and leptin).
- Results
The result of Physical activity (hour/day) was effectively 11 hour/day?
When I analyzed the “Effect of MED-02 on Body Fat Mass and Percentage” (Figure 2), I did not observe significant differences between the groups or when compared with the baseline data in each group. The authors should revise the statistical analysis.
In general, the authors should present results as data ranges in the manuscript because the SDs are greater than the Mean values.
- Discussion
The discussion should be improved.
Author Response
Response to Reviewer 2 Comments
The analysis of probiotic effects on some diseases like obesity provides scientific knowledge supporting its clinical use and, therefore, is a valuable input. The authors suggested that MED-02 could be a potential probiotic supplement to prevent obesity.
In general terms, it is an interesting study, but some questions need to be addressed:
We thank you for your constructive comments. The Reviewer’s comments and our responses to them are described below. The original comments by the reviewer are in black font; they are followed by our responses in red font. Line numbers in our responses are the line number of the revised manuscript.
Point 1: (M&M) Study Design and Treatment Materials
The authors should describe the capsule composition with the administered probiotic and the placebo in detail.
Response 1: Thank you for your suggestion. In line 83-85, We added more details of capsule composition. The 500 mg of MED-02 capsule included Limosilactobacillus fermentum MG4231 and MG4244 each at 2.5 x 109 CFU with 250 mg of maltodextrin. The placebo capsules were containing 500 mg of maltodextrin.
Point 2-1: (M&M) Assessment of daily energy intake and physical activity
The authors mentioned, “During the study period, the subjects were recommended to reduce their energy in-take by 500 kcal/day less than usual and to burn 300 kcal or more by exercising every day”.
Why did they make that recommendation? In this way, the effect of the probiotic cannot be evaluated because it could be overlapped with dietary caloric restriction.
Response 2-1: Because body fat is affected not only by the test product, but also by energy intake and exercise, both the test group and the control group are recommended to maintain the generally advised levels of energy intake and exercise for obese patients for obese patients in most clinical trials for overweight and obesity. Without these recommendations, the large variations in energy intake and exercise among study subjects would skew the study results. We point out that the recommendations for energy intake and exercise in the clinical trial are commonly used in other clinical trials for overweight and obesity [1,2].
Point 2-2: Which was the daily Kcal uptaken of the participants? Also, were participants monitored to maintain those daily calories until the end of the study?
Response 2-2: The energy intake (Kcal/day) of participants is represented in Table 1. Every participant reported their dietary and physical activities at baseline (0 week), 6 weeks, and 12 weeks (end of the study). We described more details in line 124-126.
Point 3: (M&M) Clinical Outcomes
The authors should mention the methodology for blood lipids determinations (total cholesterol, low-density lipoprotein (LDL)- cholesterol, high-density lipoprotein (HDL)-cholesterol, and triglycerides), adiponectin, and leptin).
Response 3: Thank you for your comment. In line 148-150, the methodology is added. Blood lipids were measured with enzymatic colorimetric method at Seoul Paik Hospitals and Ilsan Paik Hospital. Adipokines were measured by ELISA kits in Clinical trial sample analysis institute certified by the Ministry of Food and Drug Safety of Korea.
Point 4: (Results) The result of Physical activity (hour/day) was effectively 11 hour/day?
Response 4: Thank you for your comment. To evaluate the physical activity, participants recorded their activity with Global Physical Activity Questionnaire (GPAQ) at every visitation (baseline, 6 weeks, 12 weeks). All physical activities such as working, walking to move place, or exercising were calculated by MET value (min/week). We added more details in line 128-135. If you mean the results at 12 weeks, there was no significant difference between MED-02 and placebo groups and within groups before and after the ingestion.
Point 5: When I analyzed the “Effect of MED-02 on Body Fat Mass and Percentage” (Figure 2), I did not observe significant differences between the groups or when compared with the baseline data in each group. The authors should revise the statistical analysis.
Response 5: Thank you for your observation. We revised the statistical analysis and then corrected the manuscript to make the meaning clearer. In figure 2, A, C and E were represented the data at baseline and 12 weeks and statistical difference within baseline and 12 weeks. On the other hand, B, D and F were represented the changed data (12 weeks-baseline) and statistical difference between MED-02 and placebo groups.
Point 6: (Results) In general, the authors should present results as data ranges in the manuscript because the SDs are greater than the Mean values.
Response 6: Thank you for your comment. As your comment, we presented the ranges of data that SDs are greater than mean value in overall.
Point 7: (Discussion) The discussion should be improved.
Response 8: Thank you for your comments. As your comment, we edited the discussion in overall. We rearranged sentences and added more descriptions.